# Soil Ca_2_SiO_4_ Supplying Increases Drought Tolerance of Young Arabica Coffee Plants

**DOI:** 10.3390/plants14233666

**Published:** 2025-12-02

**Authors:** Miroslava Rakocevic, Rafael Vasconcelos Ribeiro

**Affiliations:** 1Laboratory of Crop Physiology, Department of Plant Biology, Institute of Biology, State University of Campinas (UNICAMP), Campinas 13083-862, SP, Brazil; rvr@unicamp.br; 2Laboratório de Melhoramento Genético Vegetal, Centro de Ciências e Tecnologias Agropecuárias, Setor de Fisiologia Vegetal, Universidade Estadual do Norte Fluminense, Avenida Alberto Lamego, 2000, Parque Califórnia, Campos dos Goytacazes 28013-602, RJ, Brazil

**Keywords:** carboxylation efficiency, chlorophyll, CO_2_ assimilation, photosynthetic capacity, transpiration, water-use efficiency

## Abstract

Silicon (Si) may benefit the growth and physiology of various cultivated species, especially under stress conditions. Here, we hypothesized that soil Si supplying as Ca_2_SiO_4_ would increase the drought tolerance and water use efficiency of young *Coffea arabica* L. (Arabica coffee) plants, by maintaining shoot water status and photosynthesis under low water availability. To test such a hypothesis, morphological and physiological (leaf water potential, leaf gas exchange, photochemical activity, chlorophyll content) traits of coffee plants were evaluated under varying soil Ca_2_SiO_4_ applications (0, 3000, 6000 kg ha^−1^) and water availability. The chemical composition of plant tissues was evaluated under well-watered conditions after six months of Ca_2_SiO_4_ application, with fertilized plants showing higher concentrations of Ca (leaves and roots) and B (all plant organs) as compared to plants not supplied with Ca_2_SiO_4_ (control treatment). As there were no changes in Si concentration in plant organs under Ca_2_SiO_4_ application, our data indicate that the coffee species is a Si non-accumulator, or at least the cultivar ‘Catuaí Vermelho’ evaluated herein. Additionally, the photosynthetic capacity of coffee plants increased with 6000 kg Ca_2_SiO_4_ ha^−1^ compared to the control under well-watered conditions, as given by increases in gross and net photosynthesis under light saturation, light saturation point, maximum RuBisCO carboxylation rate, maximum electron transport-dependent RuBP regeneration, and maximum rate of triose phosphate use. Such photosynthetic improvements underlined high leaf CO_2_ assimilation, transpiration, carboxylation efficiency, and chlorophyll content in plants grown under Si supplying and well-watered conditions. The negative impact of water deficit on leaf gas exchange was alleviated by Ca_2_SiO_4_ application, but the instantaneous water use efficiency was maintained as similar in both water regimes, as expected for Si non-accumulator species. Morphologically, coffee stem diameter was increased under Ca_2_SiO_4_ application, regardless of water regime. In conclusion, our data revealed that high Ca_2_SiO_4_ doses benefit coffee performance and also suggest that the use of steel slag—an industrial byproduct rich in Ca_2_SiO_4_—can be considered as a sustainable practice for residue recycling in agriculture while improving *C. arabica* growth and physiology under varying water availability.

## 1. Introduction

In soils, the presence of silicate can increase the availability of important anions for plant growth (SO_4_^2−^, NO_3_^−^, and PO_4_^3−^), facilitating their uptake by roots [1]. Additionally, silicon (Si) can enhance cation mobility and uptake of Ca^2+^, Mg^2+^, and K^+^ while reducing the uptake of Na^+^ and Cl^−^ by roots [2]. Silicon fertilization alleviates Zn deficiency symptoms in Zn-deficient soils [3], while it improves Fe acquisition and alleviates Fe deficiency symptoms in Fe-deficient soils [4]. Plants uptake Si as mono-silicic acid (H_4_SiO_4_) from the soil solution, being transported by both passive (stream of transpiration) and active ways [5] through cooperative influx and efflux Si transporters [6]. In leaves, H_4_SiO_4_ is converted into insoluble SiO_2_ through polymerization [7], leading to Si deposition that mostly occurs in the leaf cuticle and in cell walls. Silicon accumulation has been found in roots as well [8].

Increases in soil Si availability are usually accompanied by increased Si content in plant organs, which may result in increased growth and productivity, mostly investigated in grasses and in some non-grass species such as soybean, bean, tomato, strawberry, and cucumber [9]. Silicon contributes to better plant growth, yield, and quality [10], as it increases photosynthetic pigment content, net photosynthesis, transpiration rate, stomatal conductance [11], and water use efficiency [12] and modulates various physiological and metabolic processes, though its specific roles depend on plant species and environmental conditions [13]. Interestingly, Si is classified as neither a macro- nor microelement, but it is considered as a “quasi-essential” element due to benefits to plants [14]. Physically, Si strengths cell walls by forming SiO_2_ deposits, causing biosilicification [15], improving mechanical support to tissues, and providing protection against pathogens and herbivores [16,17,18].

Silicon effects might not be exclusively due to its mechanical function. In fact, Si affects plant metabolism and stimulates the production of phenolics, flavonoids, anthocyanins, lignin, callose, and phytoalexins as well as the defense enzymes [19]. As Si binds to hydroxyl groups on proteins, it changes their activities and several cellular processes, improving plant biochemical defense and plant ability to withstand harsh environmental conditions such as drought, UV-B radiation [20], heavy metal toxicity [21], salinity [2,22], heat [23], or alkaline stress [24]. Si application decreases oxidative damage in plants induced by stresses, improving the activities of essential plant antioxidant enzymes [25,26], or non-enzymatic antioxidants (carotenoids, phenols, glycine-betaine, ascorbic acid and proline) for scavenging reactive oxygen species (ROS) and keeping plant cell homeostasis [26,27,28]. Si increases the ascorbate peroxidase activity, increases the contents of total soluble sugars and the leaf relative water content, and improves the photosynthetic rates under drought [20]. In addition to ROS scavenging, Si together with plant growth-promoting bacteria is found to modulate the crosstalk among jasmonate, gibberellin, and ethylene signaling pathways, thereby up-regulating key flavonoid biosynthesis genes, inducing accumulation of isoliquiritigenin, liquiritigenin, liquiritin, or licochalcone in *Glycyrrhiza uralensis* [29]. Under drought stress, Si augments expression levels of aquaporin genes, maintaining water status and ion balance, and helping plants to recover from stress [27]. Silicon nanoparticle application has been found to restore the nutrient transporter proteins in barley roots [30]. When stressed plants are supplemented with Si, some indicators of photochemistry, such as the quantum efficiency of photosystem II (PSII), the electron transport rate, and the photochemical quenching, are enhanced, indicating that Si protects the photochemical machinery under high energetic pressure at the PSII level [31].

Silicon fertilization has gained attention due to its noncorrosive elemental nature and benefits to crop cultivation and protection in sustainable agriculture [32]. Coffee is one of the most important crops in Brazil, the largest producer and exporter of coffee beans, producing about a third of the world’s coffee [33]. Currently, our understanding of how coffee plants respond to Si supplying is limited to crop protection and nutrition. In *Coffea arabica* L. (Arabica coffee), leaf Si spraying can control the leaf rust development [34,35], brown eye spot [36], or both diseases with only a half of traditional fungicide doses [37]. Induction of defense molecules such as tannin and lignin in coffee plants sprayed with potassium silicate was noticed, reducing *Oligonychus ilicis* (acari) attack [38]. In coffee, as in various other species, Si suppresses disease and insect pests by structural changes in roots, stems, and leaves, with plants presenting improved biomass accumulation, high root activity, and uptake of nutrients [39]. However, decreases in water use, K^+^ uptake, and growth were also found in coffee plants grown in nutrient solution enriched with Si [40]. Soil fertilization with Ca_2_SiO_4_ increases soil and plant Ca and Si concentrations, with a high Si dose decreasing root biomass without any significant impact on leaf gas exchange [41]. On the other hand, Si supplying increases plant height, and its association with high N rates increases the number and length of plagiotropic nodes and root biomass in coffee plants [42]. Finally, leaf Si spraying enhances fruit ripening, berry yield, and nitrogen use efficiency in *C. arabica* trees [43], with Arabica coffee plants treated with nanosilicon particles presenting higher photochemical resistance to heat stress [23].

Here, our hypothesis was that soil Ca_2_SiO_4_ supplying would increase the drought tolerance and water use efficiency of young Arabica coffee plants, by maintaining shoot water status and then photosynthesis under low water availability. To test such a hypothesis, soil was supplied with high doses of Ca_2_SiO_4_, and we analyzed the nutritional status and photosynthetic responses of potted plants before, during, and after water withholding. From a broad perspective, we aimed to evaluate how coffee plants would respond to Ca_2_SiO_4_, a key component of steel slag. In fact, alternative initiatives for recycling steel slag should be developed while improving crop performance of various species, as carried out in wheat [44].

## 2. Results

### 2.1. Nutritional Status in Young Coffee Plants Grown Under Ca_2_SiO_4_ Application

The Si concentration in plant organs did not change with increasing soil Ca_2_SiO_4_ supplying, with roots showing the highest concentrations as compared to leaves and stems (Figure 1A). Among the macronutrients, only Ca concentration increased in both leaves and roots of young coffee plants with increasing Ca_2_SiO_4_ doses (Figure 1B). Higher concentrations of Ca, N, and P were found in leaves as compared to stems and roots (Figure 1B and Appendix A), whereas the highest concentrations of K and Mg were noticed in roots (Appendix A). Among the coffee organs, the lowest concentrations of macronutrients were found in stems (Figure 1B and Appendix A). Considering micronutrients, only B concentration increased in leaves, stems, and roots of young coffee plants grown under Ca_2_SiO_4_ application (Figure 1C). The highest concentrations of B, Cu, and Mn were found in leaves, whereas the highest Fe and Mg concentrations were noticed in roots (Figure 1C and Appendix A).

### 2.2. Photosynthesis as Affected by Ca_2_SiO_4_ Application

Plants subjected to Ca_2_SiO_4_ application showed higher photosynthetic performance, as evidenced by *A*-PPFD and *A*-*C*_i_ curves (Figure 2A,B). In fact, we found higher maximum RuBisCO carboxylation rate (*V*_cmax_), maximum electron transport-dependent RuBP regeneration (*J*_max_), maximum rate of triose phosphate use (TPU), maximum gross photosynthesis under natural CO_2_ and saturating light conditions (*A*_max___gross_), maximum photosynthesis under natural CO_2_ and saturating light conditions (*A*_max_), and light saturation point (LSP) under 6000 kg Ca_2_SiO_4_ ha^−1^ as compared to control plants (Figure 2C–F). The dark respiration (*R*_d_), light compensation point (LCP), and apparent quantum yield (Φ) were not affected by Ca_2_SiO_4_ application, varying around 0.63 ± 0.11 μmol m^−2^ s^−1^, 16 ± 3 μmol m^−2^ s^−1^, and 0.039 ± 0.022 μmol μmol^−1^, respectively.

### 2.3. Leaf Water Potential, Leaf Hydraulic Conductance, Leaf Gas Exchange, and Photochemistry Under Ca_2_SiO_4_ Application and Water Deficit

Regardless of the Ca_2_SiO_4_ application, the leaf water potential measured at pre-dawn (Ψ_L_pd_) diminished gradually up to the maximum water deficit (15th day after water withholding), when plants under water deficit (WD) presented lower Ψ_L_pd_ than well-watered plants (WW) (Figure 3A). When rewatering, Ψ_L_pd_ was fully recovered after one day. In general, Ca_2_SiO_4_ supplying did not change Ψ_L_pd_ during the water deficit period. Well-watered plants had higher leaf hydraulic conductance (*K*_L_) than plants under WD after 15 days of water withholding, regardless of the Ca_2_SiO_4_ application (Figure 3B).

Leaf CO_2_ assimilation (*A*) varied over the drought period and recovery (Figure 4A). Initially, the highest *A* values were found in plants fertilized with 3000 kg Ca_2_SiO_4_ ha^−1^, but such difference due to Ca_2_SiO_4_ applications was lost after one week under water deficit (Figure 4A). After 15 days of experiment, Ca_2_SiO_4_ supplying with 6000 kg ha^−1^ caused higher *A* values than in control plants in both water regimes, and a strong reduction in *A* was found under water deficit. While the negative effect of water deficit on *A* was offset after the first day of rewatering, the positive effects of Ca_2_SiO_4_ applications remained. On the third day of recovery, the positive Si effects were lost, and there was no full recovery of *A* in coffee plants. Interestingly, the first day of recovery was marked by high relative air humidity and the lowest leaf-to-air vapor water difference (VPD_L_, ~0.6 kPa) found during the experimental period. On the other hand, the highest VPD_L_ values were noticed at the maximum water deficit, reaching ~2.3 kPa (Appendix A).

We found a decreasing trend of stomatal conductance (*g*_s_) during the experimental period in both water regimes, regardless of Ca_2_SiO_4_ application (Figure 4B). Changes in *g*_s_ due to water supply were noticed from the eighth day and during the recovery period. On the eighth day, *g*_s_ was lower under water deficit as compared to well-watered conditions in plants under no Ca_2_SiO_4_ application. Also on the eighth day, Ca_2_SiO_4_ application caused decreases in *g*_s_ as compared to plants without Ca_2_SiO_4_ supplying and grown under well-watered conditions. At the maximum water deficit, we found no decreases in *g*_s_ of coffee plants supplied with 6000 kg Ca_2_SiO_4_ ha^−1^ under water deficit. On the first day of rewatering, the highest *g*_s_ values were measured under the highest Ca_2_SiO_4_ supplying (6000 kg ha^−1^), while the lowest *g*_s_ occurred in plants without Ca_2_SiO_4_ supplying or maintained under well-watered conditions throughout the experimental period (Figure 4B). The leaf transpiration rate (*E*) followed a temporal dynamics similar to *A* (Figure 4A,C). At the maximum water deficit, Ca_2_SiO_4_ application (3000 and 6000 kg ha^−1^) increased *E* in both water regimes, but plants facing low water availability presented the lowest *E* values (Figure 4C). After rewatering, *E* was increased, and the positive effect of Ca_2_SiO_4_ supplying was lost.

The instantaneous water use efficiency (WUE) was not changed by Ca_2_SiO_4_ application, but plants facing water deficit presented higher WUE than well-watered plants during the recovery period (Figure 5A). The intercellular CO_2_ concentration (*C*_i_) varied over the experimental period, being the highest at the maximum water deficit, and the lowest on the first day of recovery (Appendix A). While the lowest *C*_i_ was measured in plants supplied with 6000 kg Ca_2_SiO_4_ ha^−1^, the highest *C*_i_ occurred in plants without Ca_2_SiO_4_ supplying, irrespective of water regime. The instantaneous carboxylation efficiency (*CE*) showed an increasing trend with increasing Ca_2_SiO_4_ doses, with the highest *CE* measured in plants supplied with 6000 kg ha^−1^ (Figure 5B). *CE* was decreased by water deficit, and differences between water deficit and well-watered plants were increased after 15 days of water withholding. On the first day of recovery, the *CE* was similar between water regimes. However, differences between water deficit and well-watered plants appeared again on the third day of rewatering (Figure 5B), indicating a partial recovery of plants.

The maximum quantum efficiency of PSII (F_v_/F_m_) was changed by Ca_2_SiO_4_ application only at the beginning of the experiment, being lower (~0.74) in plants fertilized with 6000 kg ha^−1^ (Figure 6A). Variation in water availability did not change F_v_/F_m_ during the experimental period, varying by around 0.77. At the maximum water deficit, the leaf chlorophyll content varied due to Ca_2_SiO_4_ application, with the highest values found at 6000 kg ha^−1^ in both water regimes (Figure 6B).

### 2.4. Plant Morphology Under Ca_2_SiO_4_ Application and Water Deficit

Water deficit decreased the number of leaves, without any influence of Ca_2_SiO_4_ application (Figure 7A). Interestingly, non-significant changes were noticed in plant leaf area due to water deficit or Ca_2_SiO_4_ application (Figure 7B). At the end of the experimental period, stem height, leaf dry mass, and specific leaf mass were similar between water regimes and among Ca_2_SiO_4_ doses, being 50.4 ± 1.3 cm, 16.8 ± 1.3 g, and 70.38 ± 1.11 g m^−2^, respectively. Stem diameter increased due to Ca_2_SiO_4_ application, but it was not changed by water deficit (Figure 7C).

### 2.5. Correlation Analysis Among Variables Changed by Ca_2_SiO_4_ Supplying

Only variables that responded significantly to Ca_2_SiO_4_ application on the 15th day of experiment were considered in the correlation analysis (Figure 8). In control plants, *A* was positively correlated with *E* and *CE*, while Chl content was negatively corelated with *A*, *E*, and *CE* (Figure 8A). *C*_i_ was negatively correlated with *CE* in the control treatment and also in plants supplied with 3000 kg Ca_2_SiO_4_ ha^−1^ (Figure 8A,B, respectively), as expected due to calculation of *CE* (*A*/*C*_i_). Interestingly, in all Si treatments, *g*_s_ was positively correlated with *C*_i_ (Figure 8A–C). Under 6000 kg Ca_2_SiO_4_ ha^−1^, *A* was positively correlated with *g*_s_, *E*, *C*_i_, and *CE*, *A* positively correlated with *E*, and *E* with *C*_i_ (Figure 8C). The stem diameter (SD) was positively correlated with Chl content under 3000 kg Ca_2_SiO_4_ ha^−1^ (Figure 8B).

## 3. Discussion

Here we revealed the integrated ecophysiological responses of one-year-old coffee plants to Ca_2_SiO_4_ application, as an alternative and sustainable practice to enhance coffee performance either under well-watered conditions or under water deficit. Under adequate water availability, we found increases in leaf B and Ca accumulation and enhanced photosynthetic capacity (*V*_cmax_, *J*_max_, TPU, *A*_max_gross_, *A*_max_, and LSP) under high Ca_2_SiO_4_ supplying (Figure 1 and Figure 2). We believe that such responses to high Ca_2_SiO_4_ application supported high leaf CO_2_ assimilation (Figure 4A and Figure 5B) and increased transpiration (Figure 4C), Chl content (Figure 6B), and stem diameter (Figure 7C) found under water withholding. Additionally, as there were no changes in Si concentration in plant tissues under Ca_2_SiO_4_ supplying (Figure 1A), our data indicate that the coffee species is a Si non-accumulator, or at least the cultivar ‘Catuaí Vermelho’ evaluated herein. In fact, differential intra-specific responses to Si were already reported in cowpea (Si non-accumulator) [45] and cucumber (from accumulator to non-accumulator) [46].

In Si non-accumulator species, like tomato, *Glycyrrhiza uralensis*, and pepper, enhanced drought tolerance due to Si supplementation was associated with modifications in physio-biochemical functions (leaf photosynthesis, stomatal conductance, the efficiency of PSII, contents of proline, flavonoids, lycopene, ascorbic acid, proteins, carotenoids, activities of enzymes that scavenge ROS, suppression of chlorophyll degradation, accumulation of carbohydrates by regulating enzyme activities, and gene expression levels indirectly providing substrates for cell wall synthesis), improving plant growth and yield under drought [47,48,49]. Such responses were found even with plants lacking Si deposition in the leaf cuticle and in cell walls, which would enhance their rigidity and resistance to biotic and abiotic stresses, as reported in Si accumulators, like rice or maize [50].

The drought stress was significant after 15 days of water withholding, as indicated by large decreases in Ψ_L_pd_ and *K*_L_ in coffee plants, regardless of Ca_2_SiO_4_ application (Figure 3A). Improved water status due to Si fertilization was found in various species, even non-accumulators such as cowpea plants [45]. Some cowpea cultivars present increases in Ψ_L_ and water use efficiency (WUE) and in overall tolerance to water stress under leaf Si supplying, and such responses are mediated by increased proline concentration and antioxidant enzyme activities. In coffee plants, WUE did not change with Ca_2_SiO_4_ supplying during the experimental period, and non-significant changes in WUE were noticed even on the 15th day of water withholding, irrespective of water treatment (Figure 5A). In general, Ca_2_SiO_4_ supplying and water deficit did not improve WUE, as we initially hypothesized, because Ca_2_SiO_4_ application equally stimulated net photosynthesis (Figure 4A) and transpiration (Figure 4C). In fact, increased WUE is frequent in Si accumulators, such as maize and rice [51]. However, Si fertilization has been found to alleviate the effects of drought stress even in Si non-accumulators, such as tomato and canola, with plants showing increases in root hydraulic conductance under Si supplying [52]. Such a pattern of response was not found in coffee plants, which presented significant decreases in Ψ_L_pd_ and *K*_L_ under drought conditions even when supplied with Ca_2_SiO_4_ (Figure 3). Overall, further research on Arabica coffee is needed to elucidate the underlying mechanisms associated with Si uptake and accumulation, and to understand eventual intra-species variations in *Coffee* spp.

From a nutritional perspective, application of Si shows the potential to increase nutrient availability in the rhizosphere and root uptake, modulating the transcriptional regulation of transporters for both root acquisition and tissue nutrient homeostasis [53]. Accordingly, Si fertilization has increased the uptake and accumulation of some macro- and micronutrients, such as (i) P uptake, reinforcing the resistance against disease-causing pathogens [54], (ii) N uptake, assimilation, and remobilization under N starvation [55], and also improving N assimilation and Chl synthesis in plants under excess of nitrate [56]. In our experiment, concentrations of N and P were not modified by Ca_2_SiO_4_ application (Appendix A). Calcium was the only macronutrient changed by Ca_2_SiO_4_ supplying with plants always presenting the highest leaf and root Ca concentrations under the highest Ca_2_SiO_4_ dose (Figure 1B). Such accumulation was due to the chemical formulation of Ca_2_SiO_4_. The increased Ca accumulation in all plant organs (Figure 1B) would help to explain how Ca_2_SiO_4_ application alleviated drought stress in coffee plants. However, Ca concentration in plant organs was not the limiting factor in control plants, and any conclusion about the role of Ca in increasing drought tolerance in coffee plants supplied with Ca_2_SiO_4_ deserves further experimentation. In general, Ca has a role as secondary messenger, triggering cellular responses such as increases in antioxidant enzyme activity and osmotic regulators, modulating stomatal movement to improve WUE, and giving strength to the cell membranes and walls [57]. Considering micronutrients, Si fertilization can cause a variety of responses, depending on the fertilizer itself, environmental conditions, organs, and plant species. Under water deficit, soil or leaf Si supplying may enhance Fe, Cu, and Mn contents in some cereals, such as in maize, wheat, and rice, all Si accumulators [58,59,60], or even decrease the concentration of micronutrients, as found in oat, also a Si accumulator [61]. In horticultural crops, Si fertilization increased Fe, Mn, and Zn uptake by melon, an intermediate Si accumulator [62], and Fe, Zn, Cu, Mn, and B concentrations in cucumber (Si accumulator) fruit [63]. Leaf Si supplying reduces Fe content and increases Cu and Mn contents in potato tuber under water deficit, with no effects on Zn, B, or Si contents [64]. Contrary to the potato (Si non-accumulator) responses, we found increases in B concentration in leaves, stems, and roots of coffee plants under Ca_2_SiO_4_ supplying and well-watered conditions (Figure 1C). Boron is found in cell walls, as it modulates cell wall synthesis and structure by affecting the plasma membrane-bound proton-pumping ATPase and ion flux [65]. Here, increases in Ca and B concentrations in coffee plants due to Ca_2_SiO_4_ supplying could enhance the antioxidant metabolism and osmotic regulators, modulating stomatal movement, improved cell wall structure, and plasma membrane ion flux—a hypothesis to be further tested.

We observed that both leaf CO_2_ assimilation and transpiration (Figure 4A,C) were under stomatal control, and non-significant changes were noticed in WUE among coffee plants grown under two contrasting water regimes (Figure 5A). This occurred because Ca_2_SiO_4_ application increased both *A* and *E*, irrespective of water regime (Figure 4A,C). The reductions in water loss due to Si supplying under drought are attributed to the interaction between stomatal regulation and cuticular water permeability, mainly related to silicate crystal depositions below the leaf epidermal cells in Si accumulators [66]. As coffee plants are not Si accumulators, such silicate deposition is unlikely here. However, stomatal regulation of water loss was found in coffee plants supplied with 6000 kg Ca_2_SiO_4_ ha^−1^ (Figure 4B). ABA and proline are compounds that play a role in the movement of the stomata, and under drought stress conditions, their contents commonly increase. In soybean, an intermediate Si accumulator, proline and ABA contents decrease, whereas protein content increases as leaf Si concentration increases [67]. In drought-stressed soybean plants, Si application enhanced *E* and *g*_s_ by cumulative water uptake and transport through Si-mediated mechanisms, such as osmotic adjustment, ROS scavenging, hormonal regulation, and protein synthesis [68]. Such mechanisms are possible in coffee Arabica plants under the drought and Si applications, which can be assessed in future work.

Enhanced CO_2_ assimilation is a consequence of improved photosynthetic capacity of Arabica coffee under high Ca_2_SiO_4_ application, as given by increases in *A*_max_, *A*_max_gross_, *V*_cmax_, *J*_max_, and TPU when compared to plants without Ca_2_SiO_4_ supplying (Figure 2). Increases in *V*_cmax_ and *J*_max_ are also reported in Si accumulator cucumber plants facing biotic stress and supplied with Si [69]. Enhanced carboxylation efficiency was in fact noticed during the experimental period, with Arabica coffee showing higher *CE* when supplied with Ca_2_SiO_4_ (Figure 5B). This response to Si fertilization varies among species, such as improvements caused by Si in carboxylation of *Talisia esculenta* (non-accumulator of Si) found under water deficit [70]. Such differential sensitivity to Si fertilization among species suggests that the physiological understanding of Si benefits to plants should be dependent on species (accumulators, non-accumulators, or intermediate) and type of stressor. One could hypothesize that Si increases photosynthetic efficiency through several indirect mechanisms, especially under stress conditions, such as (i) increasing uptake of some essential micro- and macroelements relevant to photosynthesis, as happened here with Ca and B; (ii) increasing sink strength due to enhanced plant growth, which would alter gene expression and increase RuBisCO activity/abundance, as shown in rice, a Si accumulator species [71]; iii) preventing negative effects of some environmental stress on activities of photosynthetic enzymes, as found in cucumber, a Si accumulator species [56]; or (iv) improving plant tolerance to drought stress by triggering the expression of various stress-associated genes and maintaining cellular homeostasis [66].

As Chl content (Figure 6B) and stem enlargement (Figure 7C) were observed in plants under Ca_2_SiO_4_ supplying, we could argue that such responses represent physiological and morphological strategies to alleviate water stress in *C. arabica*. Preserving Chl from degradation is a likely consequence of increased ROS scavenging capacity [68]. On the other hand, a larger diameter of main stem and branches in perennial species could indicate a greater reserve of photoassimilates, which would be beneficial in limiting conditions when photosynthesis—the main source of carbohydrates—is low, as observed in fruiting citrus plants [72]. High demand for photoassimilates during fruiting can lead to branch dieback in adult coffee plants that do not have sufficient reserves to support both fruit development and vegetative growth, and a larger stem storage capacity can help to prevent this dieback [73]. After the primary root uptake—i.e., after Si is transferred from external substrate into the cortical cells via transporters and/or passive diffusion—Si is released into the transpiration stream to the shoots via xylem and then accumulates in plant stems [8]. Si directly and indirectly facilitates the synthesis of cell wall components by regulating both cell wall metabolism and non-structural carbohydrate metabolism, thus reinforcing the cell wall, enhancing its stability, and improving the drought tolerance of *Glycyrrhiza uralensis*, a Si non-accumulator [48]. In Si non-accumulator tomato plants, root lignin accumulation was associated with silicon-induced resistance to drought [74]. As in previous species, stem Si concentration did not change in coffee plants (Figure 1A), but stem diameter was increased under Ca_2_SiO_4_ application (Figure 7C) and, probably, stem biomass.

Regarding photochemistry, Ca_2_SiO_4_ application and water regimes did not change F_v_/F_m_ (Figure 6A), and we would argue that photochemical reactions did not limit CO_2_ assimilation by coffee plants. In fact, leaf Chl content was increased under high Ca_2_SiO_4_ application irrespective of water supply (Figure 6). Increases in Chl content were found previously in forages (Si accumulators) under Si fertilization [75]. The increased Chl content with Si fertilization helps to protect the chloroplast structure and avoid decreases in chlorophyll content due to stressful conditions in sorghum, a Si accumulator [76]. Silicon application can up-regulate genes responsible for Chl synthesis and down-regulate chlorophyll-degrading enzymes, promoting increased leaf Chl levels in cucumber, a Si accumulator [56]. Eventually, molecular level research could reveal specific coffee responses to Ca_2_SiO_4_ application.

We found various physiological and morphological benefits caused by Ca_2_SiO_4_ application to coffee plants, with improved performance under water deficit. Overall, Ca_2_SiO_4_ application indirectly increased the photosynthetic capacity of coffee plants and stomatal opening, without any evident improvement in water transport or use efficiency. In species that are not Si accumulators, Si fertilization primarily acts through production of secondary metabolites, such as phenolics and terpenoids, changing salicylic acid and abscisic acid signaling in plants under abiotic stresses and mitigating oxidative injuries [77]. Whether this is the case for coffee plants, more research is needed. Increased drought tolerance allied to coffee crop protection induced by Si [39] would contribute to sustainable agriculture and recycling of industrial residues rich in Ca_2_SiO_4_, such as steel slag. Modern agricultural practices must be able to face not only an increasing food demand due to increasing population and industrialization level, but also more erratic environmental conditions with more frequent and intense drought events caused by climate change.

## 4. Materials and Methods

### 4.1. Plant Material and Growth Conditions

The red-yellow latosol used in this study was collected from the 0 to 20 cm layer from Brazilian Cerrado (22°53′ S and 47°04′ W). Soil was then sieved and dried, and chemical analyses revealed: pH (CaCl_2_) 4.3; organic matter 28 g dm^−3^; 0.9 mmol_c_ K dm^−3^, 7 mmol_c_ Ca dm^−3^; 3 mmol_c_ Mg dm^−3^; 5 mmol_c_ Al dm^−3^; 38 mmol_c_ H + Al dm^−3^; sum of bases of 10.9 mmol_c_ dm^−3^; CEC of 49 mmol_c_ dm^−3^; 3 mg P dm^−3^; 14 mg S dm^−3^; 0.29 mg B dm^−3^; 2.3 mg Cu dm^−3^; 65 mg Fe dm^−3^; 3.3 mg Mn dm^−3^; 0.5 mg Zn dm^−3^; and base saturation of 22% [78]. Particle size analysis indicated a sand/silt/clay ratio of 615:55:330 g kg^−1^. Regarding the soil fertilization, we added 60 mg N, 200 mg P, and 100 mg K per kg of soil, supplied as monoammonium phosphate (MAP) and potassium chloride (KCl). In addition, dolomitic limestone was applied to increase the base saturation to 50% [79], using 1 g kg^−1^ of soil (equivalent to 2000 kg ha^−1^). This procedure was performed for all pots, considering that the main purpose was to use calcium silicate as a source of silicon (Si) and not as a corrective of soil acidity. Subsequently, fertilization was carried out with 200 mL of a micronutrient solution every 30 days, as performed by Ramos [80]. The calcium silicate treatments (here called ‘Ca_2_SiO_4_ treatments’) consisted of pure Ca_2_SiO_4_ equivalent to 3000 and 6000 kg ha^−1^. Our reference (control) treatment did not receive Ca_2_SiO_4_. Steel slag is a source of calcium silicate, and its use in agriculture is a way of recycling such industrial byproduct while promoting crop performance. Steel slag applications in coffee culture have positive impacts on nitrogen use efficiency, yield, and fruit ripening [36], while high doses up to 20000 kg ha^−1^ have been tested on sugarcane with positive results [81]. Importantly, we used Ca_2_SiO_4_ and not steel slag as a source of Si in our study.

After fertilization and Ca_2_SiO_4_ supplying, soil moisture was maintained for 15 days at 80% of field capacity. Then, six-month-old seedlings of *C. arabica* cv. ‘Catuaí Vermelho’ were transplanted into pots (20 L) containing fertilized soil and grown under greenhouse conditions, where air temperature varied between 17.6 and 34.8 °C, and the maximum PPFD was about 1200 mmol m^−2^ s^−1^. At this time, the coffee plants were about 22 cm tall, with a stem diameter around 0.3 cm, and about seven leaf pairs, mainly distributed on the orthotropic trunk, with average plant leaf area of 280 cm^2^. The soil water retention curve was evaluated using deformed samples [82], and field capacity (0.20 m^3^ m^−3^) was obtained at 30 kPa [83]. The pots were irrigated at two- to three-day intervals to maintain soil moisture close to 80% of field capacity, which was evaluated by weighing pots with a digital scale.

### 4.2. Experimental Procedure

Our experiment was conducted in two steps, first by analyzing coffee plant responses to Ca_2_SiO_4_ application, and second by evaluating the interaction between Ca_2_SiO_4_ applications and water availability. For the first step, the coffee plants were about one year old and grown for approximately six months under Si supplying and well-watered conditions. At this time, leaf, stem, and root samples were collected for nutritional analyses (Section 4.2.1). In addition, plants receiving Ca_2_SiO_4_ equivalent to 6000 kg ha^−1^ were compared to control ones in terms of photosynthetic responses to increasing light intensity (*A*-PPFD) and air CO_2_ concentration (*A*-*C*_i_), as described in Section 4.2.2. The study of photosynthetic responses to light and CO_2_ is highly time-consuming; then *A*-PPFD and *A*-*C*_i_ responses were measured only in the most contrasting treatments, assuming that such contrast will help to elucidate how Ca_2_SiO_4_ could affect *C. arabica* plant physiology. In the second step, coffee plants grown under varying Ca_2_SiO_4_ supplying (0, 3000, and 6000 kg ha^−1^) were subjected to well-watered conditions (WW) and to water deficit (WD) by water withholding. Four plants of each Ca_2_SiO_4_ treatment were kept irrigated (twelve plants in total), while another twelve plants were subjected to water deficit until significant wilting (loss of turgor) of the youngest fully expanded leaf, in the early morning. Measurements of leaf gas exchange, photochemistry, and leaf water potential were then taken during the water deficit and after re-irrigation (Section 4.2.3 and Section 4.2.4). At the end of the second step, some morphological traits were evaluated in all treatments (Section 4.2.5).

#### 4.2.1. Plant Nutritional Analyses

About six months after transplanting and growing under well-watered conditions, plant nutritional status was analyzed in root, stem, and leaf samples before water withholding, considering three plants for each treatment. Macronutrients (N, P, K, Ca, Mg, and S) and micronutrients (B, Cu, Fe, Mn, and Zn) were evaluated following Bataglia et al. [84], whereas Si was quantified according to Korndörfer et al. [85].

#### 4.2.2. Photosynthetic Responses to Light and Air CO_2_ Concentration

Photosynthetic response curves to vary the photosynthetic photon flux density (*A*-PPFD) and air CO_2_ concentration (*A*-*C*_i_) were taken in well-watered plants for evaluating the Ca_2_SiO_4_ effects on the photosynthetic machinery of coffee plants. Measurements of leaf CO_2_ assimilation (*A*) were conducted with a portable open-system IRGA (LI-6400, LI-COR, Lincoln, NE, USA) in fully expanded and sun-exposed leaves, at a leaf temperature of 26 °C and leaf-to-air vapor pressure difference (VPD_L_) lower than 1.0 kPa. In *A*-PPFD curves, PPFD varied from 1500 to 0 μmol m^−2^ s^−1^ (1500, 1200, 900, 600, 400, 200, 100, 75, 50, and 0 μmol m^−2^ s^−1^), while air CO_2_ concentration was maintained at 400 mmol mol^−1^. Measurements were taken after ~2 min at each PPFD level (when total CV was lower than 0.5%), following an initial acclimation (~20 min) to 1500 µmol m^−2^ s^−1^. *A*-PPFD curves were fitted using the nonrectangular hyperbola model [86], as shown in Equation (1):
(1)$$A=\frac{\Phi \cdot \mathrm{PPFD}+A_{\max\_\mathrm{gross}}-\sqrt{{\{⟦\Phi \cdot \mathrm{PPFD}+A_{\max\_\mathrm{gross}}⟧}^{2}-\left.4\cdot \Theta \cdot \Phi \cdot A_{\max\_\mathrm{gross}}\right\}}}{2\cdot \Theta }-R_{d}$$ where *A* is the leaf CO_2_ assimilation (μmol m^−2^ s^−1^), Φ is the apparent quantum yield (μmol μmol^−1^) calculated as the angle of the initial linear region of *A*-PPFD curves, *A*_max_gross_ is the gross photosynthesis under light saturation (μmol m^−2^ s^−1^), *R*_d_ is the dark respiration (μmol m^−2^ s^−1^), and θ is the curve convexity (dimensionless). Light compensation point (LCP, µmol m^−2^ s^−1^), maximum photosynthesis under light saturation (*A*_max_, μmol m^−2^ s^−1^, estimated as: *A*_max_ = *A*_max_gross _· 0.9 − *R*_d_), and light saturation point (LSP, µmol m^−2^ s^−1^, as PPFD level for *A*_max_) were calculated from the above-stated model.

In *A*-*C*_i_ curves, PPFD was fixed at 1000 μmol m^−2^ s^−1^ and air CO_2_ concentration (*C*_a_) was varied from 50 to 2000 μmol mol^−1^, as follows: 400, 200, 100, 75, 50, 400, 600, 800, 1000, 1500, 1700, and 2000 μmol mol^−1^. Measurements of *A* and intercellular CO_2_ concentration (*C*_i_) were taken after ~4 min in each step, when total CV was lower than 0.5%. Based on *A*-*C*_i_ curves, we estimated the maximum RuBisCO carboxylation rate (*V*_cmax_, µmol m^−2^ s^−1^), the maximum electron transport-dependent RuBP regeneration (*J*_max_, µmol m^−2^ s^−1^), and the maximum rate of triose phosphate use (TPU, µmol m^−2^ s^−1^), all corrected to 25 °C. The C_3_ photosynthesis model proposed by Farquhar et al. [87] was used to fit the *A*-*C*_i_ data, and we used the spreadsheet proposed by Sharkey [88].

#### 4.2.3. Leaf Gas Exchange and Photochemistry

The measurements of leaf gas exchange were performed alongside the water deficit and after soil re-irrigation, starting when plants were well-watered. After 15 days growth under water deficit (the maximum water deficit), leaf gas exchange was measured, and then all plants were irrigated in the late afternoon to evaluate plant recovery capacity. Measurements of *A*, *C*_i_, stomatal conductance (*g*_s_), transpiration (*E*), instantaneous carboxylation efficiency (*CE* = *A*/*C*_i_), instantaneous water use efficiency (WUE = *A*/*E*), and VPD_L_ were taken with the same equipment used for evaluating photosynthetic responses, also on fully expanded and light-exposed leaves. The measurements were taken between 9h00 and 10h30 under PPFD of 1000 μmol m^−2^ s^−1^ and natural variation in air temperature (ranging between 23.5 and 26.3 °C), air CO_2_ concentration (388 ± 11 μmol mol^−1^), and relative humidity (ranging between 34 and 66%). Again, measurements were taken under temporal stability and low total CV (<1%).

The minimum (F_0_) and maximum (Fₘ) chlorophyll fluorescence signals were measured in dark-adapted (30 min) leaves, and then the maximum quantum efficiency of PSII (F_v_/F_m_, with F_v_ = F_m_–F_o_) was estimated at the same sampling time of leaf gas exchange. Chlorophyll fluorescence measurements were taken with a PAM-2000 fluorometer (Heinz Walz GmbH, Effeltrich, Germany), using the light saturation pulse method (λ < 710 nm, PPFD ~ 10000 μmol m^−2^ s^−1^, 0.8 s). Chlorophyll (Chl) content (a + b) was evaluated in leaf discs (10 cm^2^) taken from leaves from all treatments at the maximum water deficit, following the method proposed by Lichtenthaler and Wellburn [89].

#### 4.2.4. Leaf Water Potential and Hydraulic Conductance

Leaf water potential (Ψ_L_) was measured in the early morning (pre-dawn) on the same days in which leaf gas exchange was evaluated along with the experiment. We used a pressure chamber model 3005F01 (SoilMoisture Equipment Corp, Santa Barbara, CA, USA) for measuring Ψ_L_. At the maximum water deficit, Ψ_L_ measurements were taken at both pre-dawn (Ψ_L_pd_) and 14h00 (Ψ_L_14_), and then the leaf hydraulic conductance (*K*_L_) was estimated as performed by Silva et al. [90] and shown in Equation (2):
(2)$$K_{L}=(\frac{g_{s}\times {VPD}_{L}}{{(\Psi }_{L\_pd}-\Psi_{L\_14})\times P_{atm}})$$ where P_atm_ is the atmospheric pressure (kPa).

#### 4.2.5. Morphological Measurements

At the end of the experimental period, the number of leaves was counted, the stem diameter was measured at the soil level using a digital caliper (model 100.179G, Digimess, São Paulo, SP, Brazil), plant height was evaluated on the orthotropic stem (considering the apical meristem) using a measuring tape, and the plant leaf area was measured with a digital planimeter model LI-3000 (LI-COR, Lincoln, NE, USA).

### 4.3. Statistical Analysis

All analyses were performed using the ‘R’ software, version R.4.2.1 [91]. Our experiment was in a completely random factorial design, with one to three factors, depending on the variable. When considering *A*-PPFD and *A*-*C*_i_ curves, there was only one factor: Si treatments (0 and 6000 kg ha^−1^). For morphology, Chl content and *K*_L_, there were two factors: water availability (well-watered, WW; and water deficit, WD) and Ca_2_SiO_4_ treatments (0, 3000 and 6000 kg ha^−1^). For leaf gas exchange and Ψ_L_, there were three factors: water availability (well-watered, WW; and water deficit, WD); Ca_2_SiO_4_ treatments (0, 3000, and 6000 kg ha^−1^); and sampling time (five times). For plant nutritional status, there were two factors: plant organ (leaves, stems, and roots) and Ca_2_SiO_4_ treatments (0, 3000, and 6000 kg ha^−1^). Then, two- and three-way ANOVAs were processed after the use of mixed linear modeling (lme function and maximum likelihood from ‘nlme’ package), considering water availability, Ca_2_SiO_4_ treatments, plant organ, and sampling time as fixed factor effects, while plant number (repetition) was a random effect (n = 3 or 4). If no significant interaction (starting from the most complex one, where three factors are interacting in three-way ANOVA) was found, the model reduction was applied and fitted by using the lme function, considering again all factors as fixed or random, as mentioned before. Bartlett’s homogeneity and Shapiro’s normality tests were performed for each variable. For comparing the average values estimated by ANOVA, we used the Tukey HSD and ‘lsmeans’ and ‘multcompView’ packages. The estimated means, standard errors (SE), and *p*-values are shown in the figures. The Pearson coefficient was used to correlate physiological and morphological traits at the maximum water deficit, using the complete matrix and the ‘corrplot’ package. The correlations were analyzed only for variables that showed significant responses to Ca_2_SiO_4_ applications and were performed separately for each of three Ca_2_SiO_4_ treatments. A significance level of 0.05 was used for all analyses.

## 5. Conclusions

The novelty of this work was related to integrated responses of *C. arabica* plants to Ca_2_SiO_4_ application, as an alternative sustainable practice able to increase photosynthetic capacity under well-watered conditions, and to alleviate negative impacts of water deficit on leaf gas exchange. We were able to confirm partially our initial hypothesis—soil Ca_2_SiO_4_ supplying would increase the drought tolerance and water use efficiency of young Arabica coffee plants, by maintaining shoot water status and photosynthesis under low water availability. Here, Ca_2_SiO_4_ supply increased not only CO_2_ assimilation but also transpiration, maintaining similar WUE among Si doses and water regimes. During the water deficit, higher transpiration was driven by higher stomatal conductance under Ca_2_SiO_4_ application, while higher CO_2_ assimilation was associated with enhancements in the Calvin–Benson cycle, given by increased carboxylation activity of RuBisCO and increased regeneration of RuBP. Taken together, such physiological responses alleviated the negative impact of water deficit on coffee plants, which showed higher stem diameter when fertilized with Ca_2_SiO_4_. As there were no changes in Si concentrations in plant organs due to Ca_2_SiO_4_ application, Arabica coffee—or at least cv. ‘Catuaí Vermelho’—may be classified as a Si non-accumulator. As Ca_2_SiO_4_ is the main component of steel slag, a byproduct of industrial activity. We provided evidence that steel slag can be recycled in agriculture—providing both Ca and Si—as a potential sustainable practice for *C. arabica* cultivation.

## Figures and Tables

**Figure 1 plants-14-03666-f001:**
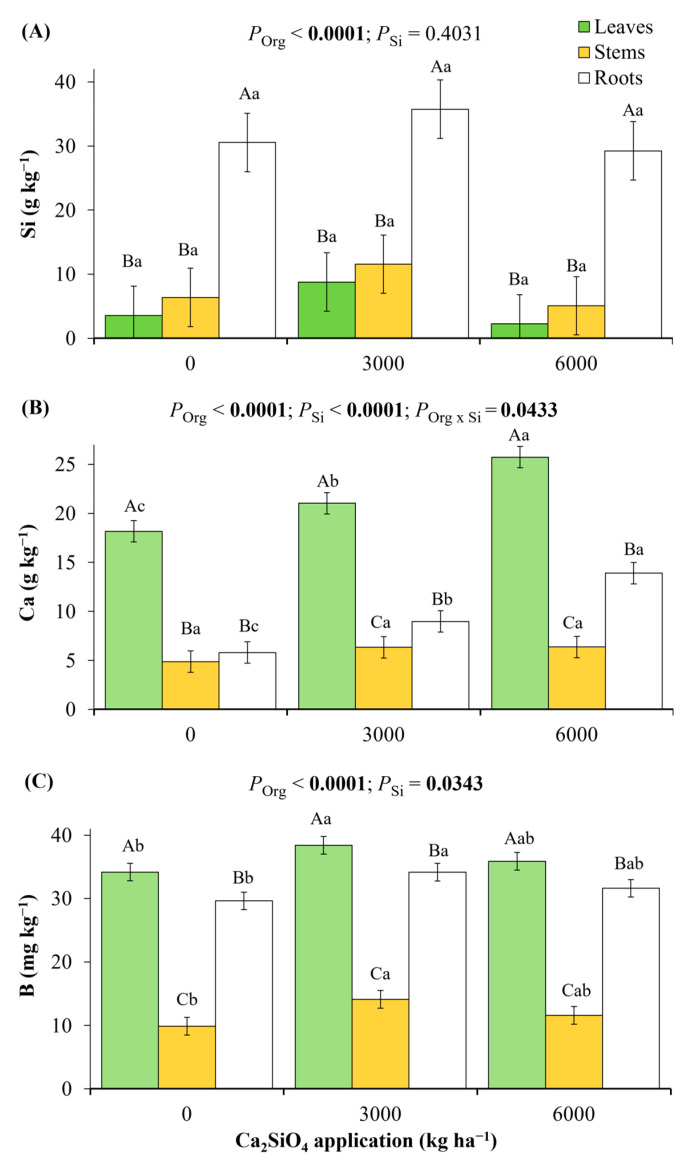
Variations in (**A**) Si, (**B**) Ca, and (**C**) B in leaves, stems, and roots of young coffee plants grown for six months under varying Ca_2_SiO_4_ application, corresponding to 0, 3000, and 6000 kg ha^−1^. Estimated mean ± SE and *p*-values (bold when significant) are shown (n = 3). Uppercase letters compare concentrations among organs (Org) for each Ca_2_SiO_4_ treatment (Si), while lowercase letters compare Ca_2_SiO_4_ treatments in each plant organ.

**Figure 2 plants-14-03666-f002:**
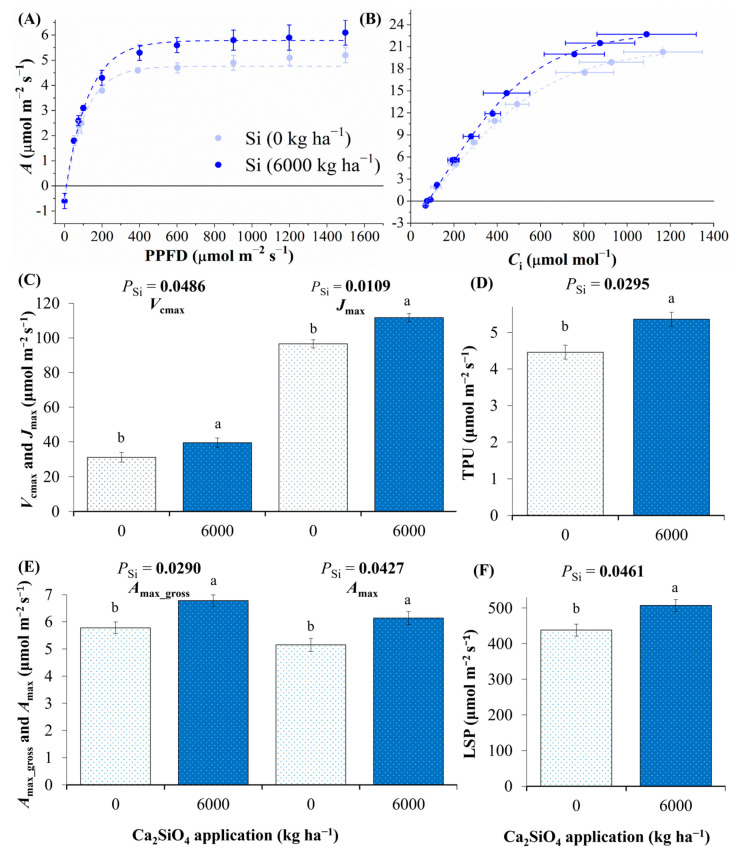
Response curves of photosynthesis to (**A**) light (*A*-PPFD) and (**B**) CO_2_ (*A*-*C*_i_), and parameters estimated from them: (**C**) maximum RuBisCO carboxylation rate (*V*_cmax_), maximum electron transport-dependent RuBP regeneration (*J*_max_); (**D**) maximum rate of triose phosphate use (TPU); (**E**) maximum gross photosynthesis under natural CO_2_ and saturating light conditions (*A*_max___gross_), and maximum photosynthesis under natural CO_2_ and saturating light conditions (*A*_max_); and (**F**) light saturation point (LSP), evaluated in well-watered coffee plants under 0 and 6000 kg Ca_2_SiO_4_ ha^−1^. Estimated mean ± SE and *p*-values (bold when significant) are shown (n = 3). Lowercase letters compare leaf responses to Ca_2_SiO_4_ applications (Si).

**Figure 3 plants-14-03666-f003:**
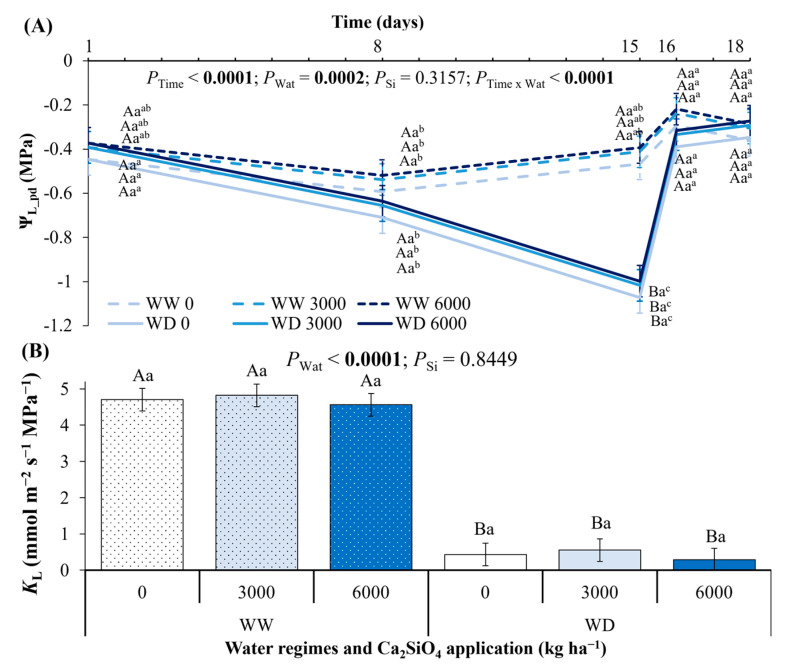
Changes in (**A**) pre-dawn leaf water potential (Ψ_L_pd_), and (**B**) leaf hydraulic conductance (*K*_L_) of young coffee plants grown under Ca_2_SiO_4_ supplying (Si), corresponding to 0, 3000, and 6000 kg Ca_2_SiO_4_ ha^−1^, and subjected to two water regimes: well-watered (WW) and water deficit (WD) for 15 days, followed by re-watering and recovery period (days 16 and 18). Estimated mean ± SE and *p*-values (bold when significant) are shown (n = 3–4). Uppercase letters compare water regimes (Wat) for each Ca_2_SiO_4_ treatment and for each day of measurements; lowercase letters compare Ca_2_SiO_4_ treatments (Si) for each water treatment and for each day of measurements; superscripted lowercase letters compare responses over time (Time) for each water regime and Ca_2_SiO_4_ treatment.

**Figure 4 plants-14-03666-f004:**
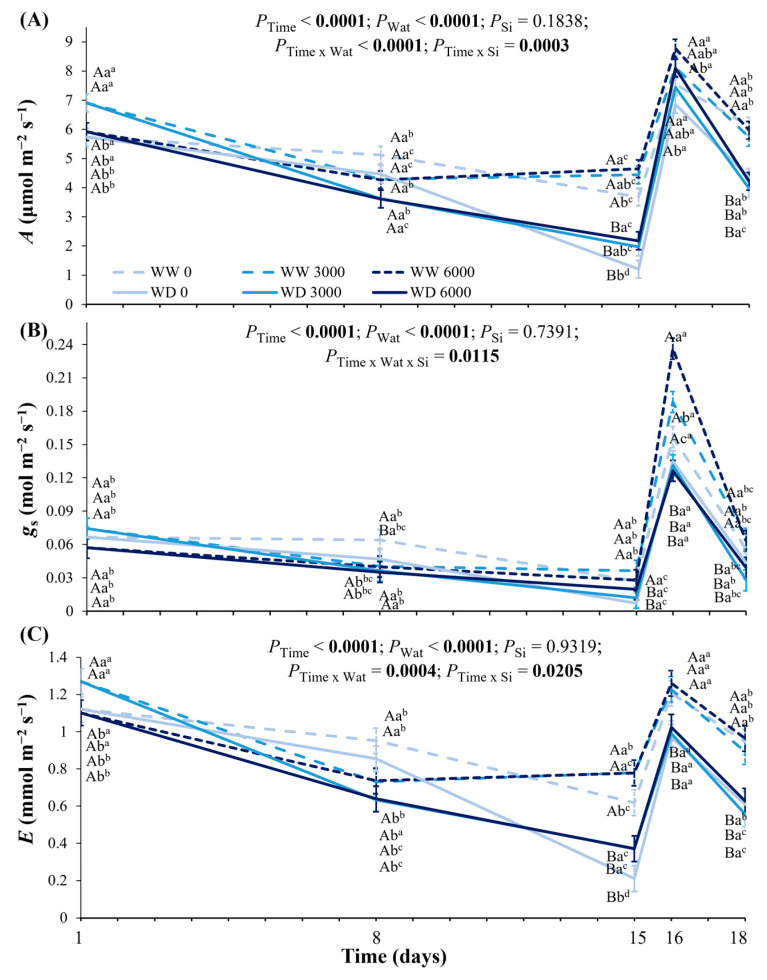
Variations in (**A**) leaf CO_2_ assimilation (*A*), **(B**) stomatal conductance (*g*_s_), and (**C**) transpiration (*E*) of young coffee plants grown under Ca_2_SiO_4_ supplying, corresponding to 0, 3000, and 6000 kg Ca_2_SiO_4_ ha^−1^ and subjected to two water regimes: well-watered (WW) and water deficit (WD) for 15 days, followed by re-watering and recovery period (days 16 and 18). Estimated mean ± SE and *p*-values (bold when significant) are shown (n = 3). Uppercase letters compare water regimes (Wat) for each Ca_2_SiO_4_ treatment and for each day of measurements; lowercase letters compare Ca_2_SiO_4_ treatments (Si) for each water treatment and for each day of measurements; superscripted lowercase letters compare responses over time (Time) for each water regime and Ca_2_SiO_4_ treatment.

**Figure 5 plants-14-03666-f005:**
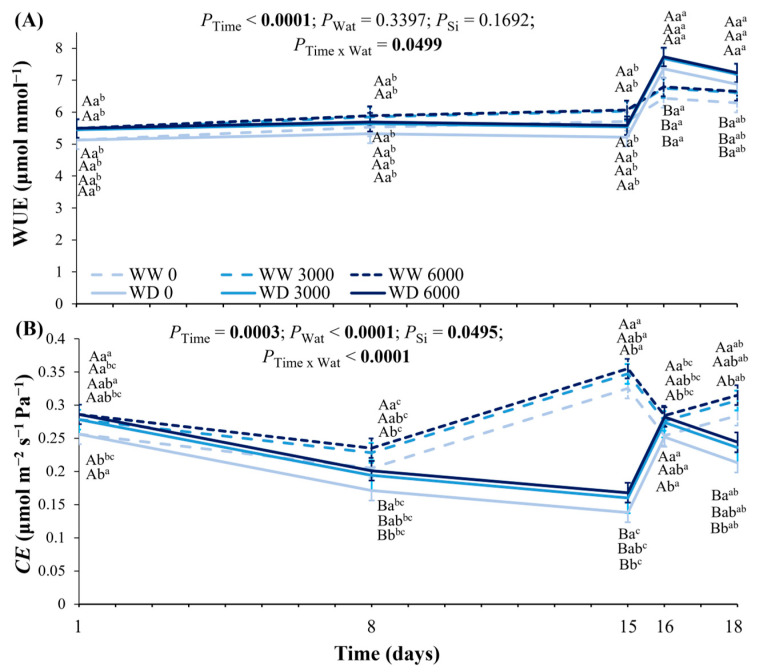
Changes in (**A**) instantaneous water use efficiency (WUE) and (**B**) instantaneous carboxylation efficiency (*CE*) of young coffee plants grown under Ca_2_SiO_4_ supplying, corresponding to 0, 3000, and 6000 kg Ca_2_SiO_4_ ha^−1^ and subjected to two water regimes: well-watered (WW) and water deficit (WD) for 15 days, followed by re-watering and recovery period (days 16 and 18). Estimated mean ± SE and *p*-values (bold when significant) are shown (n = 3). Uppercase letters compare water regimes (Wat) for each Ca_2_SiO_4_ treatment and for each day of measurements; lowercase letters compare Ca_2_SiO_4_ treatments (Si) for each water treatment and for each day of measurements; superscripted lowercase letters compare responses over time (Time) for each water and Ca_2_SiO_4_ treatment.

**Figure 6 plants-14-03666-f006:**
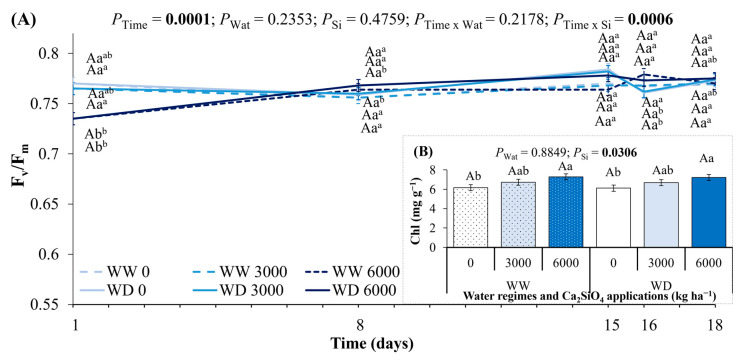
Changes in (**A**) maximum quantum efficiency of PSII (F_v/_F_m_) and (**B**) total chlorophyll (Chl) concentration in leaves of young coffee plants grown under Ca_2_SiO_4_ supplying, corresponding to 0, 3000, and 6000 kg Ca_2_SiO_4_ ha^−1^, and subjected to two water regimes: well-watered (WW) and water deficit (WD) for 15 days, followed by re-watering and recovery period (days 16 and 18). Estimated mean ± SE and *p*-values (bold when significant) are shown (n = 4). Uppercase letters compare water regimes (Wat) for each Ca_2_SiO_4_ treatment and for each day of measurements; lowercase letters compare Ca_2_SiO_4_ treatments (Si) for each water regime and for each day of measurements; superscripted lowercase letters compare responses over time (Time) for each water and Ca_2_SiO_4_ treatment.

**Figure 7 plants-14-03666-f007:**
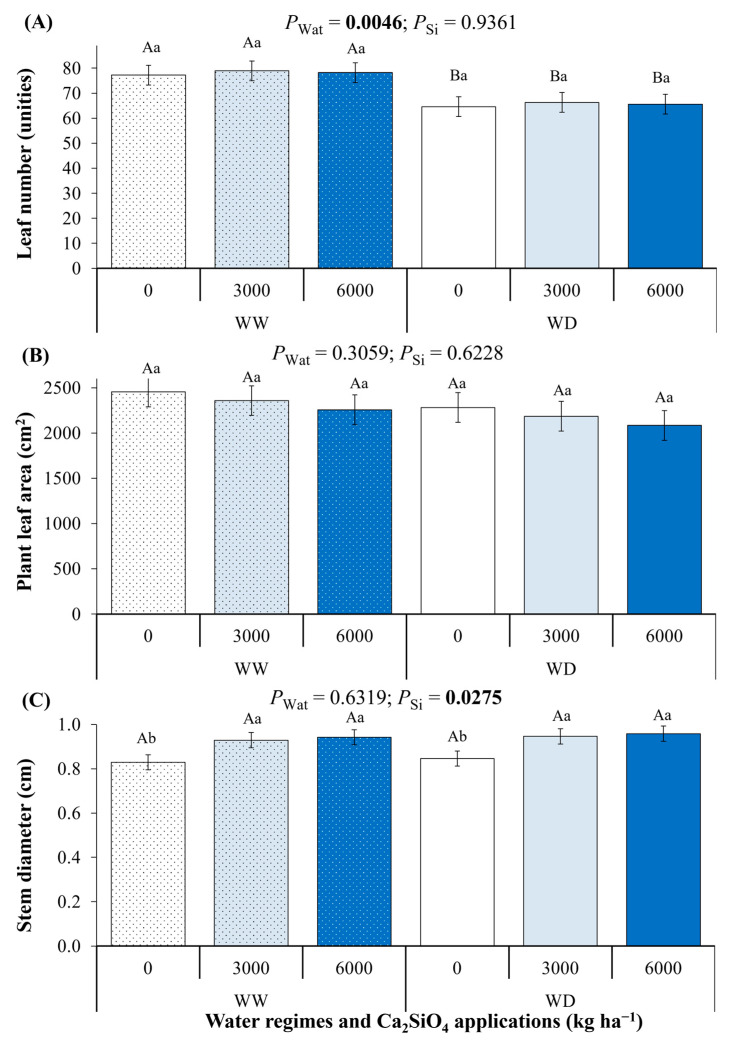
Morphological parameters: (**A**) leaf number per plant, (**B**) total leaf area per plant, and (**C**) stem diameter evaluated in young coffee plants (about one year old). Plants were grown under Ca_2_SiO_4_ supplying, corresponding to 0, 3000, and 6000 kg Ca_2_SiO_4_ ha^−1^, and subjected to two water regimes: well-watered (WW) and water deficit (WD). Estimated mean ± SE and *p*-values (bold when significant) are shown (n = 4). Uppercase letters compare water regimes (Wat) for each Ca_2_SiO_4_ treatment; lowercase letters compare Ca_2_SiO_4_ treatments (Si) for each water regime.

**Figure 8 plants-14-03666-f008:**
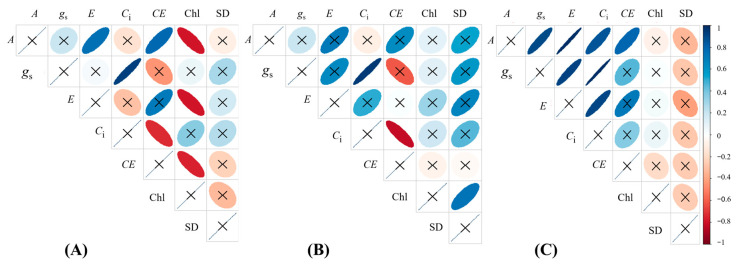
Graphical presentation of Pearson’s correlation coefficients (values corresponding to ellipse size and color intensities) with *p*-values (significant when ellipses are not crossed with "x" at < 0.05, n = 6) for correlations among leaf CO_2_ assimilation (*A*), transpiration (*E*), intercellular CO_2_ concentration (*C*_i_), instantaneous carboxylation efficiency (*CE*), leaf chlorophyll content (Chl), and stem diameter (SD) of young *Coffea arabica* plants fertilized with Ca_2_SiO_4_ at (**A**) 0 kg ha^−1^, (**B**) 3000 kg ha^−1^, and (**C**) 6000 kg ha^−1^. Data from both water regimes were pooled.

## Data Availability

The raw data supporting the conclusions of this article will be made available by the authors on request.

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
