# Peer review of "Soil Ca2SiO4 Supplying Increases Drought Tolerance of Young Arabica Coffee Plants"

_plants, 2025, doi:10.3390/plants14233666_

Round 1
Reviewer 1 Report
Comments and Suggestions for Authors
Comments and suggestions are provided in the attached file.

Author Response
The authors have addressed an important issue related to the increase of drought tolerance in Arabica Coffee Plants. The proposed alleviation method involves the soil application of silicon. This research topic is of high significance, given the increasing exposure of crops to abiotic stresses resulting from ongoing climate change. Results section provided in the manuscript is extensive, however the manuscript itself contains significant interpretative inaccuracies. In my opinion, the paper requires substantial revision before it can be reconsidered for publication.
General comments
First, the authors should note that chemical elements do not occur as separate species but as components of specific compounds, and the physicochemical properties and biological activity of those compounds largely determine their effects on plants. In this study, the authors attribute the observed effects primarily to “Si fertilization.” However, the material applied to the soil was calcium silicate (Ca₂SiO₄, steel slag), which provides both Calcium (Ca) and Silicone (Si). The results shown in Figure 1A indicate that Si content in leaves, stems, and roots did not significantly increase when both doses of steel slag (3000 or 6000 kg ha⁻¹) were applied, compared with the Si content observed for the untreated control plants. In contrast, the content of Ca in leaves and roots, and content of Boron (B) in stems, leaves, and roots, increased compared to these of untreated control plants, although the increase was not also statistically significant. This finding suggests that the physiological responses of plants observed in the study, such as changes in photosynthetic parameters and water-use efficiency, cannot be unequivocally attributed to “Si fertilization” alone. Therefore, the interpretation that “Si fertilization” alleviated drought stress is not precise. A more accurate and scientifically sound statement would be that the effects observed resulted from the application of calcium silicate (Ca₂SiO₄, steel slag), which may act through both Ca- and Si-mediated mechanisms, as it consists of both these compounds. The term “Si fertilization” should thus be replaced throughout the manuscript by “Ca₂SiO₄ application” or “steel slag amendment”.
Authors: Thank you for noting this important point. Ca₂SiO₄ application is now used along the entire manuscript, including title and figures (1, 2, 3, 6, 7, S1 and S2).
Revisions should also include changes in the Discussion section to reflect this interpretation. For instance, in lines 305–306, the authors indicated that: “Calcium was the only macronutrient changed by Si fertilization, with plants always presenting the highest leaf and root Ca concentrations under the highest Si dose,”. The expression “Si fertilization” should be replaced by “Ca₂SiO₄ application,” since the observed changes are evidently related to Ca availability rather than Si accumulation. The manuscript requires a major conceptual correction regarding the interpretation of the mechanisms involved.
Authors: Please, see our previous comment. Our Discussion is now focused on Si non-accumulators, intermediate accumulators and Si non-accumulators, as well as the potential responses to Ca and B. One important point here is that Ca was not limiting in our reference treatment and this was a key point when designing the experiment. Please, see highlights (in red) along the Discussion section.
In the paragraph related to drought stress (starting from line 321), the authors attempt to associate the increased Ca concentration in coffee organs, observed after Ca₂SiO₄ application, with improved photosynthetic performance and drought tolerance. This interpretation is only partially supported by the experimental evidence. While the increase in Ca content and the corresponding improvement in physiological parameters are well documented, attributing these effects to “Si fertilization” is not justified. The data presented do not show statistically significant increase in Si concentration in plant tissues. Therefore, the conclusion that “Si alleviates drought stress” seems not to be supported by the results.
Authors: Again, you are right and we believe this issue is now solved along the revised version. In fact, the drought stress was alleviated by Ca₂SiO₄ application. For improving our understanding about the underlying processes causing enhanced coffee performance under drought stress, we addressed some biochemical, physiological and molecular responses in Discussion (lines 287-296, 347-349, 360-364, 382-383, 396-401), as suggestion for further experimentation.
The paragraph also contains an internal logical inconsistency. At the beginning, the authors correctly acknowledge that Ca accumulation results from Ca₂SiO₄ application. However, in the next sentence, they state that this explains how Si alleviates drought stress. This represents a cause–effect misinterpretation: the cause (Ca accumulation) and the effect (enhanced drought tolerance) are validly connected, but the proposed mechanism (Si-mediated) seems to be incorrectly inferred. From a logical standpoint, the statement “The increased Ca accumulation [...] may help to explain how Si alleviates drought stress” is self-contradictory.
Authors: Thank you for this comment. It is in line with the previous ones and we have now corrected/revised the entire paper. For instance, please, see lines 329-332.
What is more, the authors reference to literature describing Si-induced drought tolerance mechanisms in Si-accumulating species such as rice and cucumber. However, Coffea arabica is recognized as a low-Si accumulator. It makes a direct extrapolation of those findings inappropriate. At the same time, no references are provided regarding the well-established role of Ca as a regulator of plant stress responses including its functions in signal transduction, stomatal regulation, and membrane stabilization. It seems that the results presented by the authors support the interpretation that the observed physiological and biochemical changes should be attributed primarily to Ca-mediated mechanisms rather than to Si-related effects.
Authors: In fact, Ca levels were high (and not limiting) in all treatments – even with Si-supplied plants presenting higher [Ca] than reference ones. So, if [Si] in plant organs does not explain the responses found herein, one may argue that the same is valid for [Ca]. Previously, we noticed decreases in root biomass in coffee plants supplied with high Ca₂SiO₄ level, but without any negative effect on leaf gas exchange under well-watered conditions (Silva et al., 2010). Based on this, one would expect higher coffee sensitivity to drought under high Ca₂SiO₄ fertilization. Interestingly, we did not find higher sensitivity to drought in Si-supplied plants. We have addressed this perspective in the Discussion (lines 287-296, 347-349, 360-364, 382-3 396-401).
As for the novelty of this study, I ask the authors to clearly specify what constitutes the innovative aspect of their work. It appears that the use of steel slag in coffee cultivation has already been reported in the literature (e.g., DOI: [10.1080/01904167.2021.2006707]). Moreover, the application of steel slag to alleviate the adverse effects of drought stress has also been previously described (e.g., DOI: [10.1007/s10343-023-00874-9]). Therefore, the potential beneficial effects on arabica coffee plants could be considered, at least indirectly, as expected rather than novel.
Authors: Thank you. One of papers was previously cited, and the second was included. We changed significantly our discussion and included various papers (from the last 3-4 years (18 new), following the suggestion of another reviewer. We slightly reconfigured the novelty and we believe it relays on the photosynthetic performance of coffee plants on leaf-area basis, which has not been reported or expected, or much described in literature.
The novelty of the present work might thus rely on the specific characteristics of the steel slag used e.g., its chemical characterization that can be linked to observed biological activity. To increase the scientific value and impact of the study, the experimental design could also be expanded by testing a wider range of doses or by comparing the efficiency of steel slags (meaning steel slugs with different characteristics). Such an approach would better justify the originality of the findings and their impact.
Authors: Thank you for these suggestions. New doses and new experimentation are not in our plans, and we do believe there is enough information here to justify publication. We did not use steel slag, rather we supplied plants with the main component of steel slag, i.e., Ca₂SiO₄. About the novelty, please, see our previous comment.
Methods
As for the section 4.1, the authors indicated that calcium silicate (Ca₂SiO₄, steel slag) was used in this study. However, there is no information on the characteristics of the material used. Since steel slag is not a chemically uniform substance but rather a complex mixture of silicates and oxides, its composition and reactivity depend strongly on the steelmaking process, cooling rate, and subsequent treatment. These factors determine e.g. the availability of Ca and Si. In other words, characteristics of steel slag used can affect its biological activity in terms of plant response. I ask the authors to include a complete description of the steel slag used in their experiments to ensure reproducibility and to allow meaningful comparison with other studies employing different steel slag sources.
Authors: The reviewer is right about steel slag composition. Here, we used Ca₂SiO₄, the main component of steel slag. In fact, this represents a first step for revealing how Ca₂SiO₄ could be benefic to coffee plants and then the importance of steel slag as an alternative resource for agriculture. We indicated that on lines 445-448 and added a short statement on lines 113-115.
The authors also indicated (line 406) that plants were grown under greenhouse conditions, but these conditions are not described.
Authors: Coffee plants were grown under greenhouse conditions, where air temperature varied between 17.6 and 34.8oC, and the maximum PAR was about 1200 mmol m-2 s-1 (lines 452-453).
As for the section 4.2., the authors indicated (lines 418-421) that: In addition, plants receiving Ca2SiO4 equivalent to 6000 kg ha-1 were compared to control ones in terms of photosynthetic responses to increasing light intensity (A-PPFD) and air CO2 concentration (A-Ci), as described in section 4.2.2. It remains unclear why only plants receiving Ca₂SiO₄ equivalent to 6000 kg ha⁻¹ were analyzed, whereas the 3000 kg ha⁻¹ treatment was omitted. This omission reduces the representativeness of the dataset. Moreover, excluding one treatment level may bias the interpretation toward the higher dose used.
Authors: As the measurements of photosynthetic response to increasing light and CO2 are time consuming, and the device availability is always limiting, we decided to make comparisons between the two most contrasting treatments here (lines 467-470). While we do agree that measurements taken in other treatments would be interesting, the responses found at 6000 kg ha⁻¹ are clear enough to support our conclusions.
In agronomic research, it is generally recommended that experiments are conducted over at least two consecutive growing seasons, allowing the identification of consistent trends and ensuring that conclusions are reliable. In the present study, the authors report results based on four replicates per variant treatment, but the experiment was performed only once. Considering the heterogeneity of steel slag materials, a second experimental cycle would have significantly strengthened the validity and reproducibility of the results. Conducting the study in more than one cycle would also help confirm whether the observed responses of coffee plants to Ca₂SiO₄ application are consistent.
Authors: As mentioned previously, we used Ca₂SiO₄ rather than steel slag and tested young coffee plants. Further validation under field conditions and several growing seasons is needed for any extrapolation or recommendation. Here, we evaluated the coffee responses to Ca₂SiO₄ under semi-controlled conditions, and our data revealed some potential of using Ca₂SiO₄ that must be further explored by using steel slag and field-grown plants.

Reviewer 2 Report
Comments and Suggestions for Authors
The manuscript investigated the effects of soil silicon supplying on the morphological and physiological traits of coffee plants under different soil water conditions. The research can provide a potential sustainable practice for C. arabica cultivation. However, the current manuscript has some drawbacks need to be addressed.
1, In the introduction section, considerable space is devoted to elaborating on the importance of silicon for plants, as well as the roles of silicon in coffee plants' responses to biotic stresses and certain abiotic stresses. However, the coverage of this part is insufficient; furthermore, there is no involvement regarding the effects of silicon on coffee plants' responses to drought stress. If there is a lack of relevant literature in this specific area, at least more attention should be paid to introducing the mechanisms by which silicon alleviates drought stress in plants (not limited to coffee plants). Besides, the reason for selecting Arabica coffee plants as experimental materials should be explained.
2, In the section 4.2, Line 418-421, why is the A-PPFD and A-Ci only measured at a concentration of 6000 kg ha-1 for comparison with the control, rather than at 3000 kg ha-1? Reasons for this need to be explained here.
3, Most of the cited literature is overly outdated. It is necessary to update the literature and increase the citation of papers published in the past three years, so as to demonstrate that the author has a good understanding of the current latest research status in this field.
Author Response
The manuscript investigated the effects of soil silicon supplying on the morphological and physiological traits of coffee plants under different soil water conditions. The research can provide a potential sustainable practice for C. arabica cultivation. However, the current manuscript has some drawbacks need to be addressed.
1, In the introduction section, considerable space is devoted to elaborating on the importance of silicon for plants, as well as the roles of silicon in coffee plants' responses to biotic stresses and certain abiotic stresses. However, the coverage of this part is insufficient; furthermore, there is no involvement regarding the effects of silicon on coffee plants' responses to drought stress. If there is a lack of relevant literature in this specific area, at least more attention should be paid to introducing the mechanisms by which silicon alleviates drought stress in plants (not limited to coffee plants). Besides, the reason for selecting Arabica coffee plants as experimental materials should be explained.
Authors: Thank you for those observations. All improvements related to your suggestions are tagged in blue, in the manuscript.
We included various statements about how Si could improve plant responses to drought, addressing the underlying mechanisms in Si intermediate and non-accumulators (excluders) in both Introduction (lines 70-73, 75-82) and Discussion (lines 287-289, 328-331, 347-349, 353-364, 381-383, 396-403) sections.
2, In the section 4.2, Line 418-421, why is the A-PPFD and A-Ci only measured at a concentration of 6000 kg ha-1 for comparison with the control, rather than at 3000 kg ha-1? Reasons for this need to be explained here.
Authors: As the measurements of photosynthetic response to increasing light and CO2 are time consuming, and the device availability is always limiting, we decided to make comparisons between the two most contrasting treatments here (lines 467-470). While we do agree with measurements taken in other treatments would be interesting, the responses found at 6000 kg ha⁻¹ are clear enough to support our conclusions.
3, Most of the cited literature is overly outdated. It is necessary to update the literature and increase the citation of papers published in the past three years, so as to demonstrate that the author has a good understanding of the current latest research status in this field.
Authors: We used many references from 2016-2025. In addition, and following your suggestion, we added 18 references more published in the last three-four years and replaced some of the oldest. The newly included references are tagged in blue, on the reference list, and also along the text.

Reviewer 3 Report
Comments and Suggestions for Authors
This manuscript presents a comprehensive investigation into the effects of soil applied calcium silicate (Ca₂SiO₄) on the physiology, morphology, and drought tolerance of young Coffea arabica plants. This study is well strucured and well written. The results are cleared and methodoloy are described in deailed.
Howeve the following points need to be adressed:
- The data clearly show that plant tissue Si concentration did not increase with fertilization ( as authors mentions in the Figure 1A), leading the authors to classify coffee as a Si "excluder." Yet, numerous physiological benefits are reported, for examples, enhanced photosynthesis, stem diameter. I think this creates a paradox: how can Si be responsible for these effects if it is not accumulated in the plant? Please justfy it.
- In abstract section the autors have mentioned that "fertilized plants showing high concentrations of Ca (leaves and roots) and B (all plant organs)." This could be misinterpreted as high absolute concentrations, while the data show a significant increaserelative to the control. Please check it and confirm the statement.
- The correlation analysis in Figure 8, is difficult to interpret due to the small n (n=3 per group). With only three data points, correlation coefficients can be unstable.Please justify it and revised.
- The incomplete recovery of CO₂ assimilation (A) after rewatering (Fig. 4A) is an interesting finding that is not sufficiently discussed. This could indicate some level of residual damage or a shift in resource allocation.
Comments on the Quality of English Language
There is no big issues, nbut autors need to recheck throught the manuscript
Author Response
This manuscript presents a comprehensive investigation into the effects of soil applied calcium silicate (Ca₂SiO₄) on the physiology, morphology, and drought tolerance of young Coffea arabica plants. This study is well strucured and well written. The results are cleared and methodoloy are described in deailed.
Authors: Thank you. All improvements related to your suggestions are tagged in green, in the manuscript.
Howeve the following points need to be adressed:
- The data clearly show that plant tissue Si concentration did not increase with fertilization ( as authors mentions in the Figure 1A), leading the authors to classify coffee as a Si "excluder." Yet, numerous physiological benefits are reported, for examples, enhanced photosynthesis, stem diameter. I think this creates a paradox: how can Si be responsible for these effects if it is not accumulated in the plant? Please justfy it.
Authors: In fact, coffee plants seem to be non-accumulators of Si, but this does not imply they are unresponsive to Si supplying. Here, we found some interesting responses to Si, even with [Si] in plant organs remaining similar across all treatments (see lines 287-289, 328-331, 347-349, 353-364, 381-383, 396-403).
- In abstract section the autors have mentioned that "fertilized plants showing high concentrations of Ca (leaves and roots) and B (all plant organs)." This could be misinterpreted as high absolute concentrations, while the data show a significant increaserelative to the control. Please check it and confirm the statement.
Authors: This was corrected to “compared to the control” (lines 25-26).
- The correlation analysis in Figure 8, is difficult to interpret due to the small n (n=3 per group). With only three data points, correlation coefficients can be unstable. Please justify it and revised.
Authors: The reviewer is right. With low number of repetitions, even strong correlations may not be statistically significant, as the confidence intervals may be wide. We recalculated the correlations with the data for each of three Si treatments (line 573), increasing n-value. Such correlations are shown in the new version of manuscript (lines 260-272).
- The incomplete recovery of CO₂ assimilation (A) after rewatering (Fig. 4A) is an interesting finding that is not sufficiently discussed. This could indicate some level of residual damage or a shift in resource allocation.
Authors: Thank you for such suggestion. Opinions existed in lines 177-182, but we did not open new discussion about such phenomena, because it was more related to environment than to essential plant response.

Round 2
Reviewer 1 Report
Comments and Suggestions for Authors
The authors addressed all issues mentioned in the review.